# Anti-Proliferative Potential of Cynaroside and Orientin—In Silico (DYRK2) and In Vitro (U87 and Caco-2) Studies

**DOI:** 10.3390/ijms242316555

**Published:** 2023-11-21

**Authors:** Lucia Camelia Pirvu, Lucia Pintilie, Adrian Albulescu, Amalia Stefaniu, Georgeta Neagu

**Affiliations:** 1Department of Pharmaceutical Biotechnologies, National Institute of Chemical Pharmaceutical R&D—ICCF Bucharest, 112 Vitan, 031299 Bucharest, Romania; astefaniu@gmail.com; 2Department of Synthesis of Bioactive Substances and Pharmaceutical Technologies, National Institute of Chemical Pharmaceutical R&D—ICCF Bucharest, 112 Vitan, 031299 Bucharest, Romania; lucia.pintilie@gmail.com; 3Department of Pharmacology, National Institute of Chemical Pharmaceutical R&D—ICCF Bucharest, 112 Vitan, 031299 Bucharest, Romania; rockady@gmail.com; 4Stefan S. Nicolau Institute of Virology, Molecular Virology Department, 285 Mihai Bravu, 030304 Bucharest, Romania

**Keywords:** luteolin-7-*O*-glucoside, luteolin-8-*C*-glucoside, molecular docking, in vitro studies, dual specificity tyrosine phosphorylation-regulated kinase 2, human glioblastoma U87, human colon carcinoma Caco-2

## Abstract

Luteolin derivates are plant compounds with multiple benefits for human health. Stability to heat and acid hydrolysis and high resistance to (auto)oxidation are other arguments for the laden interest in luteolin derivates today. The present study was designed to compare the in silico and in vitro anti-proliferative potential of two luteolin derivates, luteolin-7-*O*-glucoside/cynaroside (7-Lut) and luteolin-8-*C*-glucoside/orientin (8-Lut). In silico investigations were carried out on the molecular target, namely, the human dual specificity tyrosine phosphorylation-regulated kinase 2 (DYRK2) in association with its natural ligand, curcumin (PDB ID: 5ZTN), by CLC Drug Discovery Workbench v. 1.5.1. software and Molegro Virtual Docker (MVD) v. MVD 2019.7.0. software. In vitro studies were performed on two human tumor cell lines, glioblastoma (U87) and colon carcinoma (Caco-2), respectively. Altogether, docking studies have revealed 7-Lut and 8-Lut as effective inhibitors of DYRK2, even stronger than the native ligand curcumin; in vitro studies indicated the ability of both luteolin glucosides to inhibit the viability of both human tumor cell lines, up to 85% at 50 and 100 µg/mL, respectively; the most augmented cytotoxic and anti-proliferative effects were obtained for U87 exposed to 7-Lut (IC_50_ = 26.34 µg/mL). The results support further studies on cynaroside and orientin to create drug formulas targeting glioblastoma and colon carcinoma in humans.

## 1. Introduction

Effective methods exist for increasing the oral bioavailability of the natural compounds that naturally do not pass or reach in a very small amount the human circulatory system [1,2,3]. In this context, it is nowadays relevant to re-examine the biological effects of numerous compounds with established bioavailability limitations that have been excluded from prior screening studies.

The re-examination of the existing natural compounds is even more opportune as the in silico studies can provide extended chemical and biological information relevant to their use in new applications and formulas for human health benefits. Thus, by combining chemical-quantum computations with molecular docking analysis, it is possible to obtain data on the reactivity, stability, solubility, affinity, bioavailability, steric interactions, and positions of the active atoms and the most likely binding site. The latter can be used to infer potential effects on a molecular target of countless compounds, which cannot be obtained by in vitro studies; moreover, the toxic potential of the compounds in a series can be estimated as well as the quantum relative concentrations of the major compounds in a complex mixture. For example, in silico quantum-chemical calculations on the main phenolics found in galangal-cinnamon aqueous extract (GCAE) [4], by the corroboration of the softness values and the punctual concentrations of the 12 phenolics in the extract, together allowed the computation of the quantum relative concentrations of the main phenolics in GCAE: they were rutin (61.95%), ellagic acid (16.80%), chlorogenic acid (8.09%), cinnamic acid (6.20%), and gallic acid (2.99%); these are in fact the order of the four major compounds in GCAE; further docking studies on the 12 test phenolics indeed confirmed rutin’s efficacy at inhibiting three major molecular targets in SARS-CoV-2; therefore, it can be concluded that in silico studies are effective in estimating new applications of natural compounds.

Among potential candidates, polyphenols (particularly flavonoid compounds) are of the most interest since they are recognized to have holistic human health benefits and are also common in vegetal food [5]. For instance, flavone *C*-glycosides, flavonoid glycosides substituted at a phenyl ring, or flavonoid rhamnosides are known to be the major derivates in green plants, but they are understudied.

Regarding the interest in flavone and luteolin derivates, according to the reviewed data [6,7,8], luteolin aglycone exhibits key roles in tumor prevention, tumor initiation, tumor progress, and tumor metastasis in humans. For example, it can inhibit the activation of the carcinogens by suppressing cytochrome P450/CYP1 activity, known to play a crucial role in the internalization and metabolism of the xenobiotics and endobiotics in humans; luteolin can modulate the tumor cell proliferation, either by inhibiting the nuclear factor kappa light chain enhancer of activated B cells (NF-κB), mitogen-activated protein kinases (MAPKs), phosphatidylinositol 3-kinase/PI3K and Akt/Protein Kinase B (PI3K/Akt), cyclin-dependent kinase 2 (CDK2), or by stimulating the expression of the cell regulators cyclin-dependent kinase inhibitor 1B (p27/kip1), cyclin-dependent kinase inhibitor 1 (p21/waf1), and tumor suppressor protein p53 (p53); luteolin can modulate cell survival by suppressing the insulin-like growth factor type 1 receptor (IGF-IR), platelet-derived growth factor (PDGF), PI3K/Akt, epidermal growth factor receptor (EGFR), extracellular signal-regulated kinase (ERK), serine/threonine protein kinase C (PKC), NF-κB, and MAPKs signaling pathways but also cell apoptosis, either by upsetting DNA topoisomerases, X-linked inhibitor of apoptosis protein (XIAP), and fatty acid synthesis or by stimulating the expression of the death receptor 4 (DR5), Caspases, tumor necrosis factor receptor (Fas), apoptosis regulator proteins Bax also known as bcl-2-like protein 4, p53, and c-Jun N-terminal kinases (JNKs) modulators; luteolin can also inhibit the cell angiogenesis by epidermal growth factor (EGF), vascular endothelial growth factor (VEGFR), matrix metallopeptidase 9 (MMP-9), NF-κB, hypoxia-inducible factor 1-alpha (HIF-1α), hyaluronidase, and PI3K/Akt suppression, and even cell metastasis by suppressing tumor necrosis factor alpha (TNF-α), interleukin 6 (IL-6), matrix metallopeptidase 1 (MMP-1), focal adhesion kinase (FAK), Twist-related proteins, NF-κB, and PI3K/Akt and extracellular signal-regulated kinase (ERK) activities, which are all cytokines and cell modulators related to the spread of cancer cells in the body [6,7,8].

Furthermore, the relationship between the antitumor potential and the antioxidant activity of the plant compounds is widely accepted. Thus, apart from other phenolic compounds in green plants, the redox status of luteolin derivates appears to be mostly influenced by the microelements and ions in the cell’s micro-environment; studies showed that the antioxidant activity of luteolin aglycone is related to the punctual levels of Cu, V, Cd, and Fe ions in the medium; in the particular case of Fe ions in the medium, it was shown that luteolin acts as an antioxidant at concentrations below 50 μM and as pro-oxidant at concentrations above 100 μM [9]. Studies on the activated macrophages indicated that luteolin derivates can simultaneously scavenge the reactive oxygen species (ROS) in the medium and decrease the activity of cyclooxygenase-2/(COX-2) as well as lipopolysaccharides/LPS-activated nitric oxide production [10,11]. It is known that inflammation plays a crucial role in tumorigenesis [12,13], and clinical studies showed that the decrease in the proinflammatory interleukins IL-6 and IL-8 results in the reversion of the tumor resistance to chemotherapy [13,14]. On a related note, luteolin was shown to suppress numerous inflammatory mediators including IL-6 and IL-8 interleukins [15].

Another relevant aspect is the bioavailability of luteolin derivates in humans. The presystemic and systemic presences, along with the metabolism of plant phenolics in humans, have been a subject of interest for several decades [16,17,18,19,20]. In the case of luteolin derivates, some recent studies [20] have revealed that within the first 24 h from their ingestion, luteolin O-glycosides are degraded mainly by the intestinal *Eubacterium cellulosolvens*; the luteolin aglycones released are partly converted to glucuronide derivatives, which can further pass through the intestinal mucosa; however, in the case of luteolin C-glycosides, they appeared unchanged as they passed through the intestinal endothelium [20]. These results agree with previous data revealing the ability of *Eubacterium cellulosolvens* to cleave the glucose residue in homoorientin and isovitexin but not in their homologous orientin and vitexin or other C-glycosides found in green plants [19].

In silico docking studies on 50 phenolics [5] found in vegetal food have revealed that flavonoid bioavailability generally increases with a decreasing number of hydroxyl groups on the flavan ring; for example, flavan, flavanone, and isoflavone derivates were shown to have better bioavailability prognostic than flavonols and flavones derivates. In the case of flavones’ series, in silico results have revealed a decrease in the bioavailability from aglycone form to diglycoside form as follows: flavone aglycone > flacone- 6/8-*C*-glucosides > flavone-7-*O*-glucoside > flavone-5-*O*-glucoside> flavone-7-*O*-rutinoside.

As previously mentioned, data indicate that over 90% of the active compounds in oral drugs have oral bioavailability limitations [1].

The current practice of improving the oral bioavailability of drugs is heading either to the deviation of the biological behavior by shunting the regular metabolism of the active compound in the drug or towards the deviation of the physical properties, so as to enhance its solubility and absorption in the digestive system [1,3].

To summarize, the present in silico and in vitro investigations were designed in relation to a pair of luteolin glycosides, luteolin-7-*O*-glucoside (7-Lut), also known as cynaroside, and luteolin-8-*C*-glucoside (8-Lut), namely, orientin, with respect to their anti-proliferative potential in humans and the quality of being the head series of O and C luteolin glycosides in green plants; the two luteolin derivates tested were Sigma-Aldrich reference substances solved in 70% ethanol (1 mg/1 mL, *w*/*v*). In silico docking studies aimed to analyze their inhibitory potential against the dual specificity tyrosine phosphorylation-regulated kinase 2 (DYRK2), a molecular target known to play a key role in tumorigenesis. In vitro studies were designed on two types of human tumor cell lines, the colon carcinoma Caco-2 and glioblastoma U87. This made it possible to compare two very different cells in terms of enzymatic equipment for xenobiotics. The study design is also supported by a shared interest in gaining a deeper understanding of luteolin *O*/*C*-glycosides, while DYRK2 is a very convenient molecular target for the ligands in the series of secondary metabolites in green plants, since its native ligand in the Protein Data Bank (PDB) is a ferulic acid dimer, namely, curcumin (PDB ID: 5ZTN).

## 2. Results

### 2.1. In Silico Studies

In silico studies aimed to analyze the magnitude of the inhibitory activity of 7-Lut and 8-Lut on the dual-specificity tyrosine phosphorylation-regulated kinase 2 (DYRK2) in complex with its native ligand, namely curcumin, found in the Protein Data Bank as 5ZTN (PDB ID: 5ZTN) [21].

Table 1 and Table 2 present comparative data on the intermolecular interactions, explicitly hydrogen bonding (HB) between the native ligand curcumin docked in complex with the DYRK2 binding site, in comparison with the two test ligands, 7-Lut and 8-Lut. The docking simulations were performed by CLC Drug Discovery Workbench Software (QIAGEN, Aarhus, Denmark) and MVD Molegro Virtual Docker Software (QIAGEN, Aarhus, Denmark). The exact validation protocols were described in the authors’ previous studies [22,23].

Thus, by CLC simulation (Table 1), the two luteolin monoglucosides, 7-Lut (score −81.35) and 8-Lut (score −86.61) were shown as effective inhibitors of the molecular target DYRK2; their docking scores are close to that of the native ligand CUR A501 (score −83.92). By MVD simulation (Table 2), the efficacy of 7-Lut (score −128.51) and 8-Lut (score −126.66) in interfering with DYRK2 activity was confirmed; both compounds positioned as stronger inhibitors of 5ZTN in comparison with the native ligand curcumin (score −115.85). Former studies on DYRK2 and 5ZTN [24] indicated luteolin aglycone scores of −71.01 by CLC simulation and −104.91 by MVD simulation; therefore, luteolin aglycone is a weaker 5ZTN inhibitor than curcumin and the two luteolin glycosides as well.ijms-24-16555-t001_Table 1Table 1Docking results on the interaction between the two test molecules and the active site in DYRK2 (PDB ID: 5ZTN) by CLC Drug Discovery Workbench Software.Ligand Name/Chemical Structure *Score/RMSDAmino Acids GroupInteraction(HB)HBLength (Å)Co-crystallized **CUR A501**, CURCUMIN
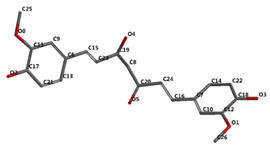
−3.92/1.99PHE 160, GLY 197, GLU 193, PHE 296, LEU 197, ASP 295, MET 226, LYS 178, VAL 163, ILE 155, ASN 280, ILE 294, ILE 212, LEU 282, PHE 228, ALA 176, ILE 155, LYS 153, LYS 165, SER 232, LEU 231, LEU 230, GLU 229, ILE 212, LEU 282, MET 233.O sp^3^ (O1)–N sp^3^ LYS 178O sp^3^ (O3)–N sp^3^ LYS 178O sp^3^(O3)–O sp^2^ GLU 1933.1022.6613.163Luteolin-7-*O*–glucoside (Cynaroside)(**7-Lut**) 
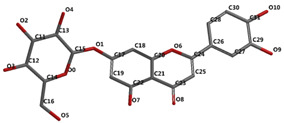
−81.35/0.28GLU 193, PFE 228, LYS 178, GLU 229, ALA 176, VAL 163, LEU 230, PHE 160, ILE 155, GLY 156, LYS 165, LYS 153, LEU 231, SER 232, MET 233, GLU 237, ASN 234, LEU 282, ILE 294, ASP 295, PHE 296, ILE 212, GLY 297O sp^3^ (O7)–O sp^2^ GLU 229O sp^3^ (O7)–N sp^2^ LEU 231O sp^3^ (O4)–O sp^2^ SER 232O sp^3^ (O10)–O sp^3^ ASP 295O sp^3^ (O10)–O sp^2^ ASP 295O sp^3^ (O10)–N sp^3^ LYS 178O sp^3^ (O9)–N sp^3^ LYS 178O sp^3^ (O9)–O sp^2^ GLU 1932.9082.7973.0923.3652.7872.7103.0422.990Luteolin-8-*C*-glucoside (Orientin)(**8-Lut**)
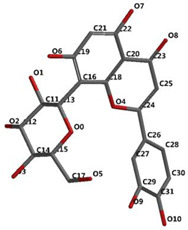
−86.61/0.06LEU 197, GLU 193, PHE 296, GLY 292, PHE 160, ASP 295, ILE 294, VAL 293, ASN 280, GLU 279, ASN 234, LEU 262, MET 233, SER 232, LEU 231, ILE 155, GLY 156, LYS 157, ILE 212, VAL 163, LEU 230, ALA 176, GLU 229, LYS 178, PHE 228.O sp^2^ (O8)–N sp^2^ LEU 231O sp^3^ (O7)–N sp^2^ LEU 231O sp^3^ (O7)–O sp^2^ LEU 231O sp^3^ (O6)–O sp^2^ ILE 155O sp^3^ (O2)–O sp^2^ GLU 279O sp^3^ (O2)–N sp^2^ ASN234O sp^3^(O10)–N sp^3^ LYS 178O sp^3^(O10)–O sp^2^ GLU 193O sp^3^ (O9)–N sp^2^ ASP 2953.0392.6233.3142.6422.5573.1303.2183.0642.936* Ligands’ structures were retrieved from the PubChem database [25] and submitted to minimization using Molecular Mechanics Force Field (MMFF) [26] using Spartan v. 14 software (Wavefunction, Inc., Irvine, CA, USA); atoms’ labeling was randomly chosen by the software.
ijms-24-16555-t002_Table 2Table 2Docking results on the interaction between the two test molecules and the active site in DYRK2 (PDB ID: 5ZTN) by MVD Molegro Virtual Docker Software.LigandMol Dock ScoreMoleculeContributionsHBHBLengthÅStericInteractionsDistanceÅ**CUR A501****Curcumin**−115.85ALA 176, ASP 295, GLU 193, GLU 229, GLY 297, ILE 155, ILE 212, ILE 294, LEU 197, LEU 230, LEU 231, LEU 282, LYS 153, LYS 165, LYS 178, MET 233, PHE 160, PHE 228, PPHE 296, SER 232, VAL 163O sp^23^(O1)–N sp^3^ LYS 178O sp^3^ (O3)–N sp^3^ LYS 178O sp^3^ (O3)–O sp^3^GLU 1933.1022.6612.249C sp^2^ (C22)–Nsp^2^ GLU 193C sp^2^ (C18)–Nsp^2^ GLU 193O sp^3^ (O3)–C sp^2^ GLU 193O sp^2^ (O3)–C sp^3^ LYS 178C sp^3^ (C15)–O sp^2^ LEU 231C sp^3^ (C23)–N sp^2^ LEU 2313.123.053.033.132.802.74**7-LUT**−128.51ALA 176, ASP 295, GLU 193, GLU 229, GLY 297, ILE 155, ILE 212, ILE 294, LEU 230, LEU 231, LEU 282, LYS 165, LYS 178, MET 233, PHE 160, PHE 228, PHE 296, SER 232, VAL 163O sp^3^ (O7)–O sp^2^ GLU 229O sp^3^ (O7)–N sp^2^ LEU 231O sp^3^ (O4)–O sp^2^ ILE 155O sp^3^ (O10)–O sp^3^ASP 295O sp^3^ (O10)–N sp^3^ LYS 178O sp^3^(O10)–O sp^3^ GLU 1933.0562.5003.3393.1372.4703.014O sp^2^ (O8)–O sp^2^ GLU 229O sp^2^ (O2)–C sp^3^ ILE 212O sp^3^ (O7)–C sp^3^ LEU 230O sp^3^ (O10)–C sp^3^ LYS 1782.292.963.083.05**8-LUT**−126.66ALA 176, ASN 234, ASN 280, ASP 295, GLU 193, GLU 229, GLU 279, GLY 297, ILE 155, ILE 212, ILE 294, LEU 230, LEU 231, LEU 282, LYS 178, MET 233, PHE 160, PHE 228, PHE 296, SER 232, VAL 163O sp^2^ (O8)–N sp^2^ LEU 231O sp^3^ (O7)–N sp^2^ LEU 231O sp^3^ (O6)–O sp^2^ ILE 155O sp^3^ (O2)–O sp^2^ GLU 279O sp^3^ (O2)–N sp^2^ ASN 234O sp^3^(O10)–N sp^3^ LYS 178O sp^3^(O10)–Osp^2^ GLU 1932.8302.5782.6582.5682.8303.3963.013O sp^2^ (O8)–Osp^2^ GLU 229O sp^3^ (O9)–C sp^3^ LYS 178C sp^2^ (C18)–C sp^3^ LEU 2822.432.983.20

Figure 1A,B, Figure 2A,B and Figure 3A,B depict the hydrogen bonding, the docking poses of the native ligand curcumin, and the two luteolin derivates interacting with the amino acid residues within the DYRK2 binding site 5ZTN, as a result of CLC simulation.

Figure 4, Figure 5, Figure 6 and Figure 7 present more detailed aspects of the interaction between the native ligand curcumin and the active pocket of DYRK2, as given from MVD simulations.

Figure 8, Figure 9, Figure 10, Figure 11, Figure 12 and Figure 13 present other aspects of the molecular interactions between the two test luteolin derivates and the amino acid residues within the binding site of the DYRK2, obtained by MVD simulation.

Table 3 presents the quantum-chemical parameters of 7-Lut and 8-Lut in comparison with luteolin, calculated by Spartan v. 20 software (Wavefunction, Inc., Irvine, CA, USA), using the Density functional theory (DFT) with the Becke’s three parameter hybrid exchange functional with the Lee–Yang–Parr correlation functional (B3LYP)-6-311(d,p) method [27] for the equilibrium geometry, in vacuum, at ground state.

Figure 14 shows the HOMO–LUMO energy diagram and Δ*E* for the two test luteolin derivates, 7-Lut and 8-Lut, predicted using the DFT/B3LYP-6-311(d,p) method.

According to the data in Table 3, it can be observed that the quantum reactivity parameters of the two luteolin derivates in comparison with their aglycone luteolin are situated at lower values, and cynaroside is of lower values than orientin. Overall, these data suggest a greater stability of 7-Lut versus a greater reactivity of 8-Lut. As the energy of the HOMO orbital (*E*_HOMO_), in all cases, is significantly smaller than the energy of the LUMO orbital (*E*_LUMO_), this fact suggests that the HOMO orbital can most easily donate electrons to form new bonds and, therefore, be involved in oxidation reactions; the LUMO orbital is the most able to receive electrons and, therefore, more likely to be involved in reduction reactions. The assessment of the energy difference between the HOMO and LUMO orbitals (Δ*E*) provides information about the chemical reactivity and kinetic stability of the ligands. According to Δ*E* values, the reactivity decreases in the following order: Lut > 8-Lut > 7-Lut. Therefore, once again, orientin (8-Lut) appears to be more susceptible to reactions than cynaroside (7-Lut) and, by comparison, to the aglycone luteolin (Figure 14).

Table 4 presents summary data of the predicted drug-likeness parameters for the two luteolin derivates in comparison with the DYRK2 native ligand, curcumin, by CLC software.

As it has been established, the drug-likeness parameters provide information about the solubility of the test molecules, their transport characteristics, and likely bioavailability in humans [27]. Among these parameters, the water-1-octanol logarithm coefficient (logP) provides the most useful information, the partition and the lipophilicity of a compound in a medium, respectively. Particularly, it was stated that a drug targeting the central nervous system should have a logP value close to 2; a drug targeting the common oral-intestinal absorption should have a logP value in the interval from 1.35 to 1.8, while a drug intended for sub-lingual absorption should have a logP value higher than 5 [28]. Also, a compound with a negative logP value is assumed to have a higher affinity for the aqueous phase than the lipid phase (i.e., the compound is more hydrophilic); logP = 0 means equal affinity for the lipid and the aqueous phases, while a positive value for logP indicates a higher affinity for the lipid phase than the aqueous phase (i.e., the compound is more lipophilic); logP = 1 means the ratio of the compound partitioning between the organic and aqueous phases is 10:1 [28].

Applied to the data series in Table 4, the water-1-octanol logarithm coefficient for 7-Lut (logP = 1.57) suggests a compound with a good bioavailability at the level of the gastro-intestinal system, which is better than the reference compound curcumin (logP = 3); 8-Lut has been computed with a logP = −0.7, indicating an affinity for the aqueous medium and high hydrophilic properties; therefore, 8-Lut appears to be a good candidate for the oral-intestinal drug formulation as well.

Furthermore, when also considering Lipinski’s violation criteria [29], the result is that the two luteolin derivates exhibit similar formulation limitations; therefore, their encapsulation might be the best option for obtaining maximum human health benefits.

### 2.2. In Vitro Pharmacological Studies

The pharmacological in vitro studies aimed to compare the cytotoxicity and anti-proliferative potential of 7-Lut and 8-Lut on two types of human cancer cell lines, glioblastoma U87 and colon carcinoma Caco-2, respectively.

In vitro studies followed the Promega protocol [30], the MTS (3-(4,5- dimethylthiazol-2-yl)-5-(3-carboxymethoxyphenyl)-2-(4-sulfophenyl)-2H-tetrazolium) assay for cytotoxicity and anti-proliferative activity studies, respectively. 7-Lut and 8-Lut were each prepared as two dilution series (*n* = 4): the first dilution series, tested on the human glioblastoma cell line U87, was prepared as 50, 25, 10, and 5 µg test compound per 1 mL sample; the second dilution series, tested on the human colon carcinoma cell line Caco-2, was prepared as 100, 50, 25, and 10 µg test compound per 1 mL sample. In parallel, the negative control series, representing the untreated U87 and Caco-2 cell lines, and the positive control series, representing the two cell lines, U87 and Caco-2, treated with the solvent sample (70% ethanol) at identical dilution series, were also prepared. The two test cell lines were cultured on EMEM (Eagle’s minimum essential medium) with 1% Penicillin-Streptomycin-Neomycin (PSN) antibiotic mixture, supplemented with 10% heat-inactivated Fetal bovine serum (FBS) in the case of U87, and 20% heat-inactivated FBS the case of the Caco-2 cell line, respectively. Briefly, by following the Promega MTS protocols for cytotoxicity and anti-proliferative assays (as described in Section 4.2. In vitro assay), the Optical Density (O.D. at 492 nm) of the test series versus the control negative series and the test series versus the control positive series were measured. The obtained results can be computed as O.D. values along the dilution series (high O.D. values suggest high cell viability) or as cell viability percentages (%) along the dilution series by reporting the O.D. of the test samples to the O.D. of the control (negative/positive) samples.

Accordingly, Figure 15A–D and Figure 16A–D show the results of the anti-proliferative experiments and the dynamics of the U87 and Caco-2 cell lines viability after treatment with 8-Lut and 7-Lut, at 24 and 48 h, respectively.

The results of the cytotoxicity experiments are presented as IC_50_, mentioned in the section dedicated to the comparison of the biological effects of 8-Lut and 7-Lut in the two in vitro experiments (Figure 17 and Figure 18).

The Optical Density values (O.D. at 492 nm) and the viability percentages (%) along the dilution series (*n* = 3) were computed for their statistical significance (Student “*t*” test); the notation (*) means results without statistical significance and *p* > 0.05, the notation (**) means results with statistical significance and 0.05 < *p* < 0.01, and the notation (***) means results with statistical significance and *p* < 0.01.

#### 2.2.1. Anti-Proliferative Potential of 7-Lut and 8-Lut on U87 Human Glioblastoma Cells

Figure 15A–D shows in vitro anti-proliferative activity of the two flavone derivates, 7-Lut (cynaroside) and 8-Lut (orientin), tested on the human tumor glial cell line U87, with four dilution series (50, 25, 10, and 5 µg/mL samples), after 24 and 48 h of cell exposure, respectively.

Overall, by comparison with the negative control series, the computed results have suggested the high efficacy of both luteolin derivates on U87, even from the very first 24 h of exposure of the cells (Figure 15A,C); at maximum concentration level in the study (50 µg/mL) the two luteolin derivates presented an inhibitory potential from 81 (24 h) to 85% (48 h) (Figure 15B,D), while an exponential trend-line of the cell viability along the dilution series was noticed. Also, the amounted inhibitory effects along the series were at higher intensity in the case of 7-Lut, while in the interval from 5 to 10 µg/mL the tumor glial cells seem to oppose to the anti-proliferative effect of 8-Lut; these might be explained by distinctive logP and polarities of the two luteolin derivates.

#### 2.2.2. Anti-Proliferative Potential of 7-Lut and 8-Lut on Colon Carcinoma Cell Line Caco-2

Figure 16A–D shows the in vitro anti-proliferative activity of 7-Lut (cynaroside) and 8-Lut (orientin) tested on human tumor colon cancer Caco-2, with four dilution series (100, 50, 25, and 10 µg/mL samples), after 24 and 48 h of exposure, respectively.

The computed results in the experiment using the human colon carcinoma cell line Caco-2 (Figure 16) and O.D. values along the dilution series (10–100 µg/mL) have revealed Caco-2 cells’ particular response to cynaroside and orientin; overall, the effects of both luteolin glycosides at 24 h (Figure 16A,C) were of a lower intensity than on U87; particularly, the anti-proliferative activity of 8-Lut and 7-Lut at 50 µg/mL on Caco-2 cells have been computed at −38% and −31% cell viability inhibitory percentages (Figure 16B,D). After 48 h of Caco-2 cell exposure to luteolin derivates, the inhibitory effects at 50 µg/mL sample raised up to −78% and −81%, which are similar to that registered on U87. Also, at the maximum concentration level in the Caco-2 study, 100 µg/mL, 7-Lut and 8-Lut lead to −86% and −85% cell viability inhibitory effects; therefore, the level of 50 µg per 1 mL sample represents the limit up to which most of the luteolin monoglycosides’ inhibitory power is developed.

By comparing the effects of the two luteolin monoglycosides in the two in vitro experiments, the result of a higher anti-proliferative efficacy in the case of the human glioblastoma U87 cell line was reached (see Figure 17 and Figure 18). It must be noticed that the prior MTS cytotoxicity assays indicated the following IC_50_ values for the two luteolin derivates: 30.91 µg/mL for 8-Lut and 26.34 µg/mL for 7-Lut in the case of U87, and 85.58 µg/mL for 8-Lut and 97.06 µg/mL for 7-Lut in the case of Caco-2.

Altogether, at the maximum effective concentration in the study, 50 µg/mL, after 48 h of exposure, 7-Lut and 8-Lut reached an anti-proliferative potential of 83–85% in the case of U87, and of 77–81% in the case of Caco-2; at 25 µg/mL, 7-Lut and 8-Lut reached an anti-proliferative potential of 75–76% in the case of U87, and of 59–61% in the case of Caco-2; these results suggest that cynaroside and orientin have a higher efficacy against the human tumor glial cell line U87 than the human tumor colon cell line Caco-2. By also comparing the IC_50_ values of the two luteolin derivates on the two cell lines, U87 and Caco-2, 2–3 times lower in the case of Caco-2 cells, we conclude that both luteolin derivates appear as very effective guardians against tumorigenesis in intestines as well.

## 3. Discussion

Luteolin derivates are secondary plant metabolites in the category of polyphenols, class of flavonoids, and sub-class of flavones. Flavones are characterized by four hydroxyl groups in the positions 3′, 4′, 5, and 7. Flavones can also occur as *O*-glycosides, if they are substituted in positions 5 and 7, or as *C*-glycosides, if they are substituted in positions 6 and 8. Therefore, apart from other flavonoid sub-classes, the subclass of flavones offers a distinct series of flavonoid compounds, i.e., of *C*-glycosides. Their role in the plant kingdom, and even more interestingly, in the human body, is little known. However, it can be speculated that they could be related to the hydrolysis’ resistance. The attribute <luteolin paradox> refers to the high resistance to temperature, acidic environments, as well as oxidative stress; therefore, luteolin derivates are good candidates for systematic studies.

In summary, scientific data [16,17,18,19,20] have proved that once ingested, flavonoid *O*-glycosides are hydrolyzed to their aglycone forms, by gastric acid and intestinal α- and β-glucosidase enzymes’ activity. Beyond that, the aglycones pass through the intestinal barrier. In the specific case of luteolin aglycone, the most effective intestinal absorption areas were found at the level of duodenum and jejunum segments and less at the level of the colon and ileum; experiments on rats also revealed the possibility of passive absorption of luteolin aglycone through the cells in the small intestine [31]. However, luteolin-*C*-glycosides were not affected by the gastric acid and small intestine enzymes in rats [20]; also, in vivo studies of rats indicated the punctual presence of luteolin-8-*C*-glucoside (orientin) in the liver, lung, and kidney tissues but not in the brain tissue [32].

After that, the cytochrome P450 monooxygenases turn the flavonoid aglycones into phase I metabolism derivatives; the derived oxidation products are further used as substrates for the next glucuronidation, sulphation, and methylation processes in phase II metabolism, by the action of the conjugating enzymes, namely, uridine-5′-diphosphate-glucuronosyltransferases, sulfotransferases, and catechol-*O*- methyltransferases [33,34,35,36].

According to the reported data, glucuronide conjugates are the most abundant luteolin metabolites in the plasma and urine of humans [37]. Also, it was found that luteolin is mainly excreted in urine, while luteolin glycosides are mainly excreted in feces [38].

Furthermore, by an in vitro comparative study on Caco-2 cells exposed to pure luteolin, pure luteolin-7-*O*-glucoside, and several aqueous extracts from *Artemisia afra*, it was shown that in terms of luteolin glucuronide production, the *Artemisia afra* aqueous extracts were more efficient than the two pure compounds, luteolin and 7-Lut [39]; in this way, luteolin-3′-*O*-glucuronide was found to be the main metabolite in plasma, followed by luteolin-4′-*O*-glucuronide and luteolin-7-*O*-glucuronide. Equally important, among the glucuronide derivatives studied, luteolin-7-*O*-glucuronide has been shown to be the most effective compound at reducing the expression of the interleukins IL-6, IL-1β, of nuclear factor NF-κB1, of chemokines C-C motif ligand 2, 3 and 5 type (Ccl2, Ccl3, Ccl5) and of transcription factor JunB, acting as proinflammatory mediators [40,41].

Comparative in vitro studies [20] on isoquercitrin, orientin, and their aglycones quercetin and luteolin also revealed that, by in vitro digestion, if the initial antioxidant activities of the two aglycones were very high, in the following intestinal phase, they decreased more sharply than the glucoside derivates; by in vivo digestion, while in plasma the antioxidant activity of the isoquercitrin (flavonol-*O*-glucoside) was more elevated than that of orientin (flavone-*C*-glucoside), in urine, the antioxidant activity of *C*-glycoside orientin was higher than that of *O*-glycoside isoquercitrin. These results confirmed that orientin remained unchanged after digestion in humans; therefore, luteolin-8-*C*-glucoside interacts as such with cells in tissues.

Finally, it is important to know the main contributors of luteolin derivates in vegetal food. After a comprehensive data comparison [42,43,44,45,46,47,48,49,50,51,52,53,54,55,56,57,58,59,60,61], it was found that peppermint, celery, Mexican oregano, tansy leaf, fenugreek seeds, rosemary, sage, and rooibos are the richest sources of luteolin derivatives in human food. Also, it was found that luteolin-*O*-glycosides are more prevalent than luteolin-*C*-glycoside.

According to the available data reports [43,44,45,46,47,48,49,50,51,52,53,54,55,56,57,58,59,60,61], the most abundant sources of luteolin-7-*O*-glucoside are Mexican oregano (297.67 mg/100 g dried herb), celery (80 mg/100 g raw seed and 1.17 mg/100 g fresh leaves), black olives (14.50 mg/100 g raw fruits), peppermint (10.90 mg/100 g dried herb), and lentils (0.12 mg/100 g raw beans).

In the case of luteolin-8-*C*-glucoside, quantitative data are scarce; according to the data reported [43,44,45,46,47,48,49,50,51,52,53,54,55,56,57,58,59,60,61], the richest sources of luteolin-*C*-glycosides are the rooibos tea (172–591 mg/dry weight) followed by the green tea (2.5–21.9 mg/dry weight), oolong tea (14.8 mg/dry weight), and black tea (5.9–14 mg/dry weight).

Another aspect that resulted from the data above is that common verbena, lemon verbena, globe artichoke, and red lettuce also contain luteolin-7-*O*-glucuronide derivates, which actually are the flavonoid derivatives found in the circulatory system, after their successive small intestine and liver metabolism in humans [62,63].

Overall, it is expected that luteolin and luteolin-*O*-glycosides are the most effective on the intestinal cells. The major argument is that they come into direct contact with them; also, luteolin was proven to inhibit the TNF-alpha-induced proinflammatory gene expression in the murine intestinal epithelial cells [64] and the human colon epithelial cells [65]. These data sustain the high usefulness of luteolin-*O*-glycosides in fighting against inflammatory bowel disease (IBD) in humans. On the other hand, luteolin-8-*C*-glucoside was found unchanged in liver, lung, and kidney tissues in humans. Together, these results likely suggest cooperation between luteolin and luteolin-*O*/*C*-glycosides in obtaining systemic activity in the human body. But, in fact, all natural luteolin derivatives end as luteolin and luteolin-8-*C*-monoglycoside (e.g., 8-Lut) after digestion in humans, since luteolin-6-*C*-glycosides (e.g., homoorientin and derivates) are cleaved by the *Eubacterium cellulosolvens* in microbiota. Therefore, these are the main arguments for the selection of 7-Lut and 8-Lut in the present study.

The selection of the Caco-2 and U87 human tumor cell lines is sustained by the opportunity to study the biological response of two types of cells, which normally interact or, on the contrary, do not interact with luteolin glycosides. Therefore, their comparative response could form the basis for further study approaches.

The selection of the molecular target, namely, dual-specificity tyrosine kinase 2 (DYRK2), is based on the numerous data pointing towards its ability to orchestrate a variety of cellular responses, linking genomic stability, tumor cell growth, tumor cell apoptosis, tumor angiogenesis, and tumor cell metabolism as well [66,67,68,69,70]. According to the reviewed data [66], a clear association between DYRK2 expression and mutations and the poor prognosis in cancer patients was established; for example, the process of phosphorylation and methylation of DYRK2 was different in normal and tumor tissues but also in the tumor microenvironment, while the expression of DYRK2 in cells was correlated with the cancer-associated fibroblast infiltration process.

This way, in silico docking studies on DYRK2 in combination with the native ligand curcumin (PDB ID: 5ZTN), using CLC and MVD software simulations, indicated a similar or a superior efficacy of 7-Lut (cynaroside) and 8-Lut (orientin) in comparison with the native ligand curcumin; therefore, they are both potentially strong inhibitors of DYRK2.

The inhibition occurs through creating hydrogen bonds (intermolecular interactions) between oxygen (O9 and O10) from -OH groups in the phenyl ring (B) of luteolin derivates and the active amino acids in 5ZTN, punctually Lysine (LYS) in the position 178 and Glutamic acid (GLU) in the position 193, as resulted from the CUR A501 study [21].

Below are presented the magnified structures of the two luteolin derivates (Figure 19A,B), as found in the PubChem database [25], for the identification of the active hydroxyls and oxygen (O9 and O10) atoms indicated by CLC and MVD simulations.

In vitro studies on the two human tumor cell lines, glioblastoma U87 and colon carcinoma Caco-2, indicated good efficacy overall of both luteolin derivates on both cell lines; the maximum anti-proliferative effects of 7-Lut and 8-Lut, computed at 81 to 85% cell viability inhibitory effects, were seated more rapidly (24 h) and at a halved concentration (50 µg/mL) in the case of U87; at an identical concentration, 50 µg/mL, after 48 h of cell exposure, 7-Lut and 8-Lut achieved 77–81% cell viability inhibitory effects in the case of Caco-2 cells.

By also considering the prior cytotoxicity results on the two test cell lines (specifically, IC_50_ values of 30.91 µg/mL and 85.58 µg/mL for 8-Lut on U87 and Caco-2, versus IC_50_ values of 26.34 µg/mL and 97.06 µg/mL for 7-Lut on U87 and Caco-2, respectively), these results have revealed the differences on the cytotoxicity of luteolin O and C glycosides on the intestinal cells in comparison with other cells less effective as equipment for xenobiotics or with which they do not naturally come into contact.

## 4. Materials and Methods

### 4.1. In Silico Assay

An in silico molecular docking study was conducted to predict the binding mode and affinity against the protein receptor dual-specificity tyrosine-regulated kinase 2, DYRK2, [21] of the two luteolin glucosides in comparison with the native ligand curcumin (PDB ID: 5ZTN). Molecular docking simulations were performed using CLC Drug Discovery Workbench Software, v. MVD 2019.7.0 (QIAGEN, Aarhus, Denmark), and MVD Molegro Virtual Docker Software, v. 1.5.1. (QIAGEN, Aarhus, Denmark) protocols, as previously described [22,23].

Overall, molecular docking allows for the establishment of an accurate prediction of the optimized conformation of tested compounds (as ligands) and their target receptor protein to achieve a stable complex. Thus, the score and hydrogen bonds formed with the amino acid residues from the binding site are further used to predict the binding modes, the binding affinities, and the orientation of the docked ligands in the active site of the protein receptor, DYRK2, respectively. In this work, the structures of the 2 + 1 test compounds were imported from the PubChem database [25] and prepared by energy minimization with Spartan v. 14 software (Wavefunction, Inc., Irvine, CA, USA) [27] using a molecular mechanics force field (MMFF) [26]. The co-crystallized curcumin was taken as a reference compound to compare the docking results of the other two test compounds, 7-Lut and 8-Lut. The calculations were completed on the optimized structures of molecules, thus presenting the configuration of minimum energy and, accordingly, an optimized geometry, in vacuum conditions without any solvent corrections. The usual steps were followed in the docking computation: ligands and protein preparation, removal of co-factors and water molecules; the setup binding site and binding pocket; docking simulations on co-crystallized and investigated ligands; validation, collecting property data and docking results. Interactions by hydrogen bonds of ligands within the active binding site of the studied protein target were identified and measured. The results are given in terms of the docking scores, which are functions of the software used in the study; particularly CLC and MVD. A drug-likeness analysis based on the structural parameters correlated with Lipinski’s rule of five [71], on luteolin and 7-Lut and 8-Lut, has also been completed. Taking into account the calculated *E_HOMO_* and *E_LUMO_* energies, the assessment of quantum-chemical parameters was completed by applying the relationships stated by Koopmans’s theorem [72].

### 4.2. In Vitro Assay

In Vitro studies were performed by the MTS assay following the protocol of the CellTiter 96AQueous One Solution Cell Proliferation AssayPromega Corporation (Madison, WI, USA) [29]; the MTS assay by Promega is a colorimetric test based on the selective ability of the viable cells in culture to reduce the tetrazolium component of [3-(4,5-dimethylthiazol-2-yl)-5-(3-carboxymethoxyphenyl)-2-(4-sulfophenyl)-2H-tetrazolium] in the medium to purple-colored formazan crystals, which can be further measured at 492 nm. The difference between the two types of MTS assays, cytotoxicity setup and anti-proliferative setup, is that in the cytotoxicity study, the cells are exposed to the test and control samples at the time when a “semiconfluent” cell culture is achieved (meaning about 70% cell proliferation), while in the anti-proliferative study, the application of the test and control samples are completed at the time when a “sub-confluent” cell culture is achieved (meaning about 30% cell proliferation).

Briefly, the two test reference compounds (7-Lut and 8-Lut) were prepared, each one as two dilution series in culture medium, starting from a stock solution of 1 mg/mL in 70% ethanol (*v*/*v*); the two-dilution series were as follows: 50, 25, 10, and 5 µg test compound per 1 mL sample in the case of U87 experiments and 100, 50, 25, and 10 µg test compound per 1 mL sample in the case of Caco-2 experiments. In parallel, following an identical algorithm, four dilution series of the solvent sample, 70% ethanol, have also been prepared (the positive control series). The cell lines investigated were the human tumor brain cell line U87 MG(ATCC-HTB-14) and the human colorectal carcinoma Caco-2 cells (ATCC, HTB-37). Briefly, after reaching the cell confluence in the study (close to 30% for the anti-proliferative MTS setting study), the cells were detached from the flask with Trypsin-EDTA. The cell suspension was further centrifuged at 2000 rpm for 5 min and, after that, re-suspended in the growth medium: EMEM (Eagle’s minimum essential medium) with 1% Penicillin-Streptomycin-Neomycin (PSN) antibiotic mixture and 10% heat-inactivated FBS for U87 MG cell line and 20% heat-inactivated FBS, required for the growth of Caco-2 cells; after that, the cells were seeded in 96-well plates at a density of 4000 cells per well in 200 μL of the culture medium. Each test sample and corresponding positive control sample in the dilution series described were applied to the cell cultures—triplicate series.

After 20 and 44 h of cell exposure to test (7-Lut and 8-Lut dilution series) and control (70% ethanol solvent) samples, the culture medium was removed. The cells were incubated with MTS solution for another two hours and, after that, the viability of the adherent cells was determined by evaluating the absorbance of solution at 492 nm (BMR-100 Microplate Reader, Boeco, Germany).

The results, Optical Density (O.D. at 490 nm) values, and the viability percentages along the series (*n* = 3) were computed for their statistical significance (Student “*t*” test); the notation (*) means results without statistical significance and *p* > 0.05, the notation (**) means results with statistical significance and 0.05 < *p* < 0.01, and the notation (***) means results with statistical significance and *p* < 0.01.

The cell culture reagents EMEM (Eagle’s minimum essential medium) and Penicillin–Streptomycin–Neomycin (PSN) Antibiotic Mixture were purchased from Thermo Fisher (Waltham, MA, USA) Distributor in Romania, and U87 and Caco-2 Growth Medium with Supplements and the FBS from Merck (Sigma-Aldrich, Saint Louis, MO, USA) Distributor in Romania.

### 4.3. Test Compounds Preparation

The two test compounds, 7-Lut and 8-Lut, and reference substances were purchased from Sigma-Aldrich (Saint Louis, MO, USA) Distributor in Romania; they were prepared as *stock solutions* of 1 mg/mL in 70% ethanol (*w*/*v*). The dilution series for in vitro studies were completed by their combination with the specific culture medium: EMEM (Eagle’s minimum essential medium) with 1% Penicillin-Streptomycin-Neomycin (PSN) antibiotic mixture and 10% heat-inactivated FBS for the U87 MG cell line and 20% heat-inactivated FBS for Caco-2 cells.

## 5. Conclusions

Based on the literature data presented here, we conclude that an augmented holistic luteolin-based product can be obtained by combining luteolin or luteolin-7-*O*-glucoside with luteolin-8-*C*-glucoside, plus optimal quantities of microelements in the series copper/Cu, vanadium/V, cadmium/Cd, and iron/Fe ions and also platinum/Pt; this combination, especially in an encapsulated form, would lead to improved bioavailability, high antioxidant activity, and also sustained anti-proliferative potential, which could be truly useful after chemical treatment of cancer tumors.

The results listed in the present study indicated that, concerning the docking analysis, 7-Lut and 8-Lut are strong inhibitors of DYRK2, more effective than the native ligand, namely, curcumin; from the in vitro side, 7-Lut and 8-Lut also were shown to be effective inhibitors of the viability of the human tumor cell line glioblastoma U87 and the colon carcinoma cell line Caco-2; the highest anti-proliferative efficacy (up to 85% cell viability inhibition) was noticed for U87 exposed to 7-Lut, while cytotoxicity experiments indicated an IC_50_ of 26.34 µg/mL; by also considering the IC_50_ cytotoxicity values from 2 to 3 times lower in the case of Caco-2 cells, both luteolin derivates appear to be very effective guardians against tumorigenesis in the intestinal cells of humans. Therefore, the cumulative results support further studies on cynaroside and orientin to create drug formulas targeting glioblastoma and colon carcinoma in humans.

## Figures and Tables

**Figure 1 ijms-24-16555-f001:**
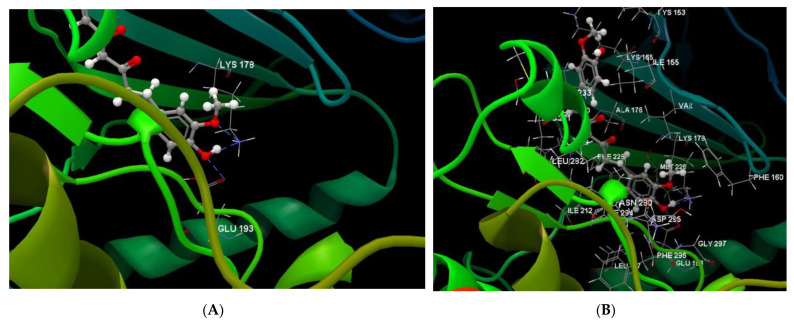
(**A**) The hydrogen bonds between the co-crystallized ligand CUR A501 and the amino acid residues within the binding site of 5ZTN. (**B**) Docking pose of the co-crystallized native ligand CUR A501 interacting with the amino acid residues within the binding site of 5ZTN.

**Figure 2 ijms-24-16555-f002:**
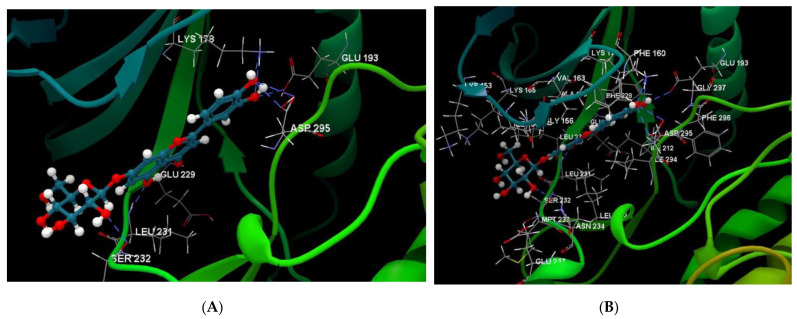
(**A**) The hydrogen bonds between 7-Lut and the amino acid residues within the binding site of 5ZTN. (**B**) Docking pose of the 7-Lut interacting with the amino acid residues within the binding site of 5ZTN.

**Figure 3 ijms-24-16555-f003:**
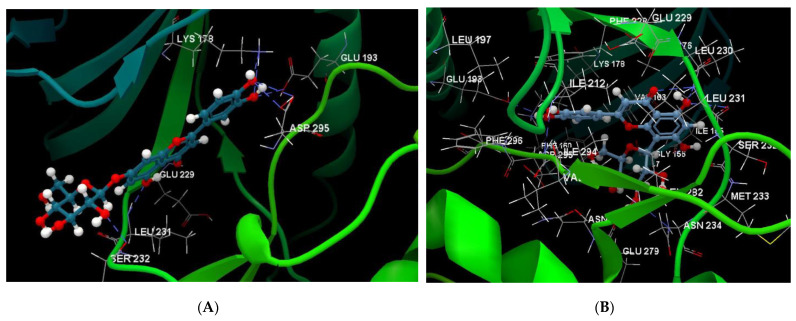
(**A**) The hydrogen bonds between 8-Lut and the amino acid residues within the binding site of 5ZTN. (**B**) Docking pose of the 8-Lut interacting with the amino acid residues within the binding site of 5ZTN.

**Figure 4 ijms-24-16555-f004:**
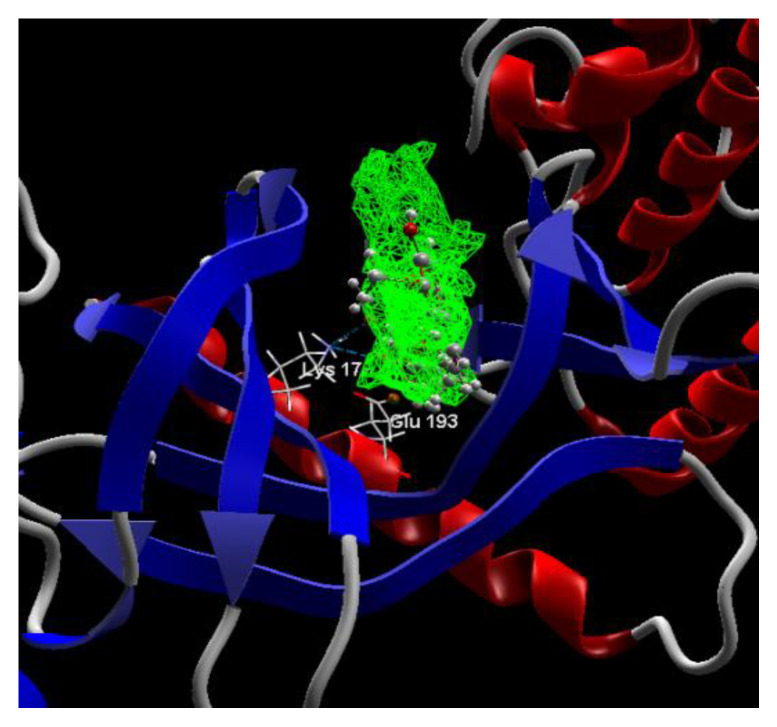
The cavity of the co-crystallized CUR A501 in the molecular target DYRK2.

**Figure 5 ijms-24-16555-f005:**
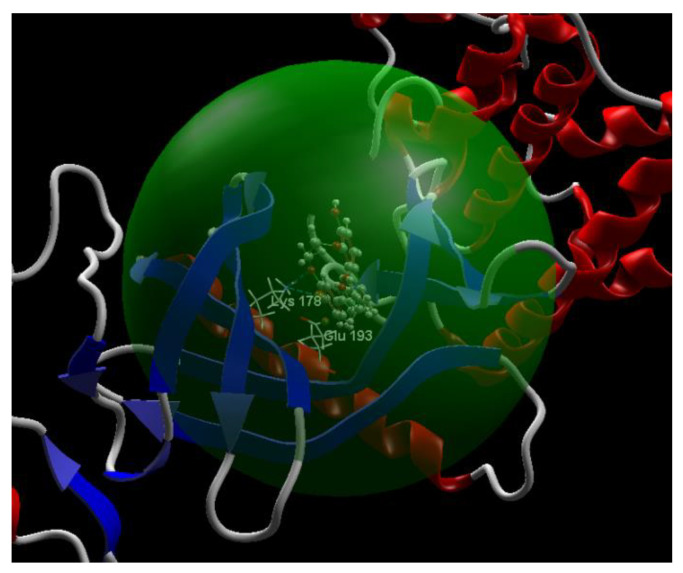
The binding site of the CUR A501 in the active pocket of the DYRK2, 5ZTN.

**Figure 6 ijms-24-16555-f006:**
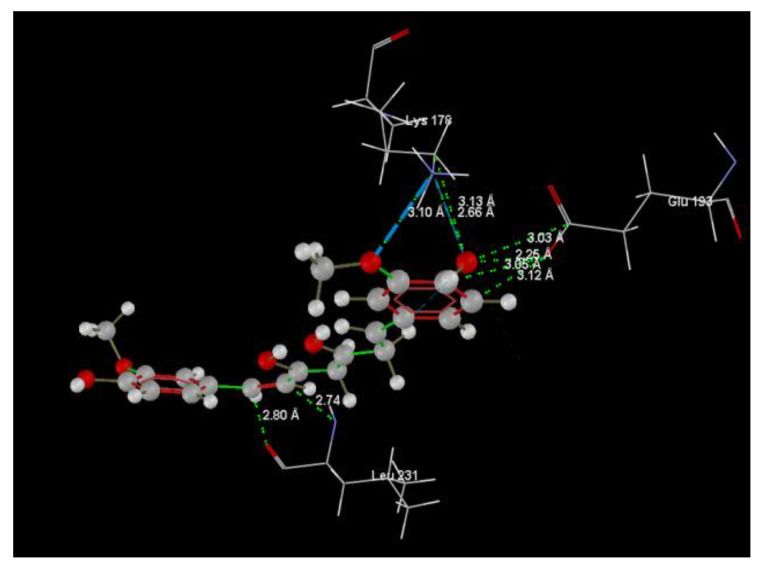
Hydrogen bonding (blue) and steric interactions (green) created between the co-crystallized CUR A501 and the 5ZTN receptor protein (2D).

**Figure 7 ijms-24-16555-f007:**
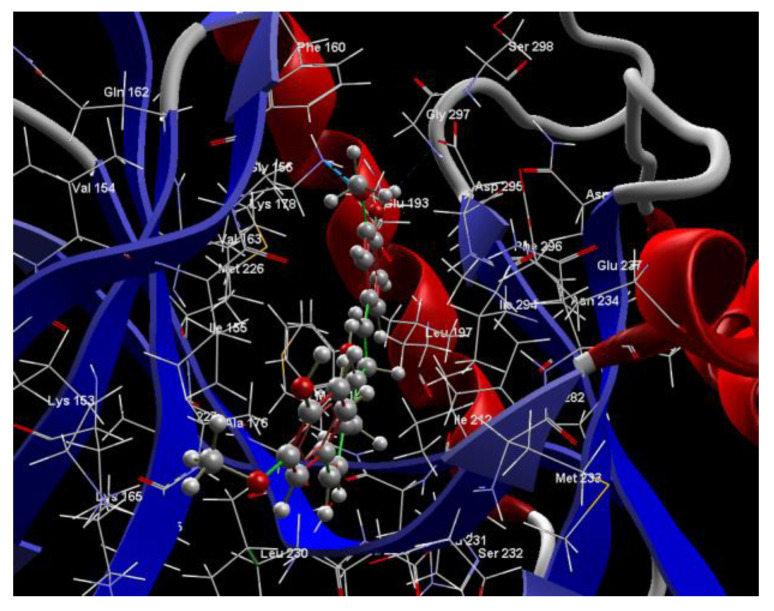
The interaction group for the co-crystallized CUR A501 and amino acid residues from the active site of 5ZTN receptor protein.

**Figure 8 ijms-24-16555-f008:**
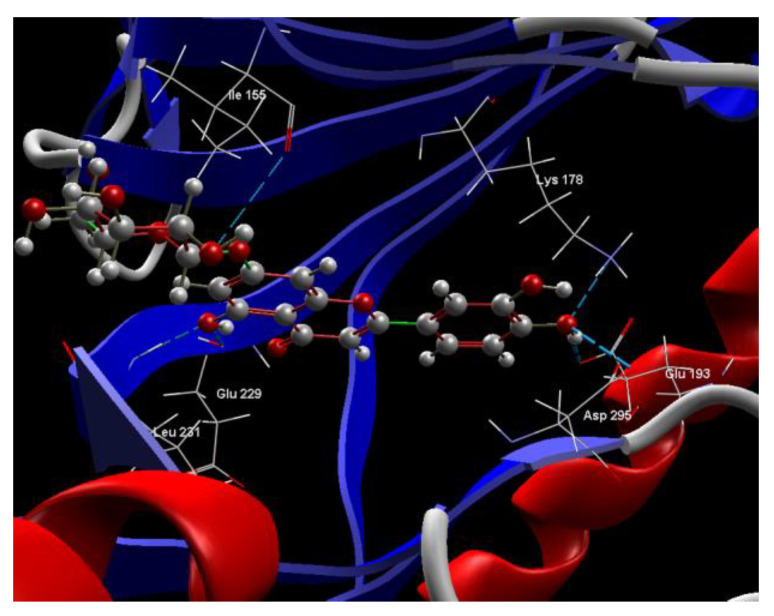
Hydrogen bonding (blue) between 7-Lut and 5ZTN binding site of the receptor protein (2D).

**Figure 9 ijms-24-16555-f009:**
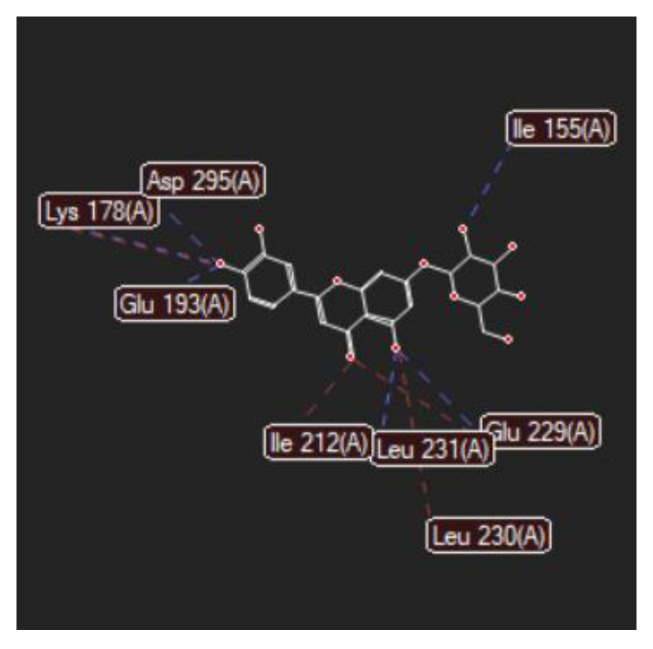
Hydrogen bonding (blue) and steric interactions (red) occurring between 7-Lut and 5ZTN binding sites of the receptor protein.

**Figure 10 ijms-24-16555-f010:**
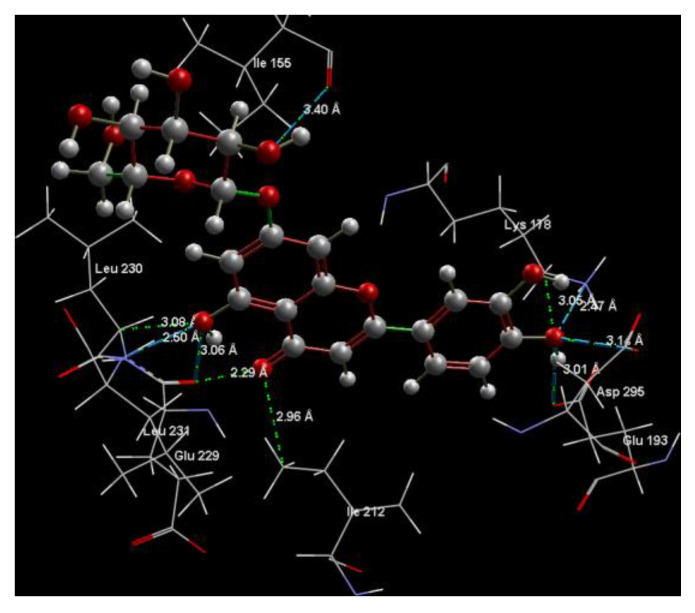
Annotations for the length of the hydrogen bonds (blue) and the steric interactions (green) occurring between 7-Lut and the amino acid residues in the binding site of 5ZTN.

**Figure 11 ijms-24-16555-f011:**
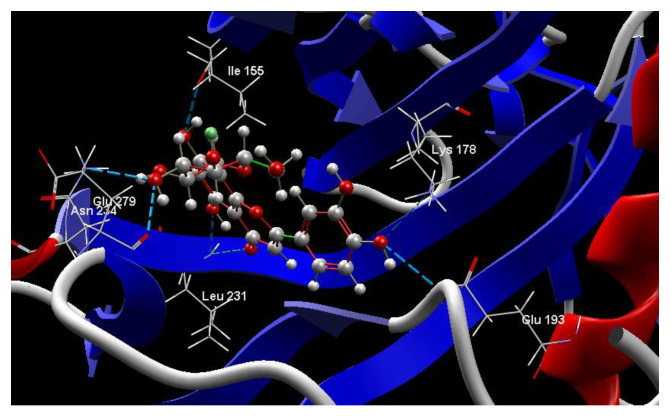
Hydrogen bonding (blue) between 8-Lut and 5ZTN binding site of the receptor protein (2D).

**Figure 12 ijms-24-16555-f012:**
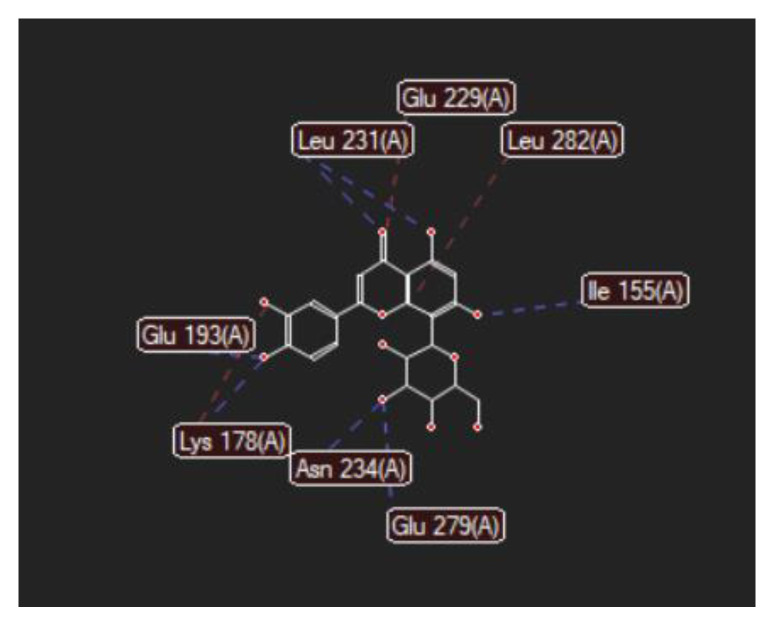
Hydrogen bonding (blue) and steric interactions (red) created between 8-Lut and 5ZTN binding sites of the receptor protein.

**Figure 13 ijms-24-16555-f013:**
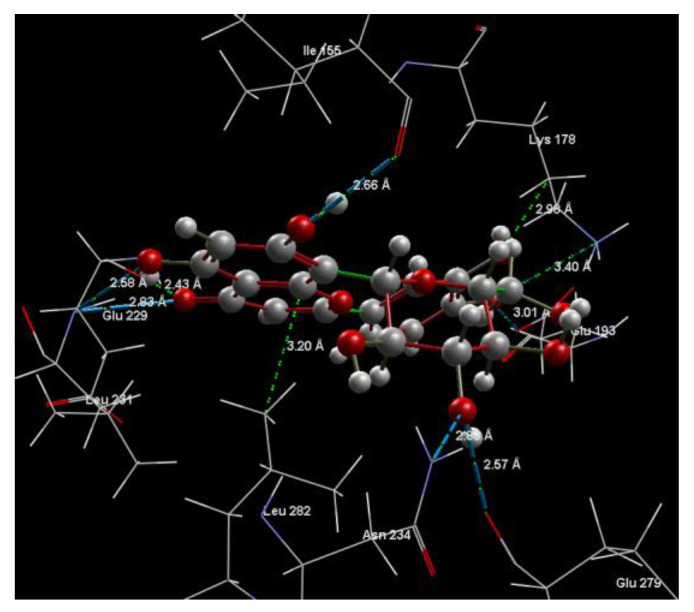
Annotations for the length of the hydrogen bonds (blue) and the steric interactions (green) occurring between 8-Lut and the amino acid residues within the binding site of 5ZTN.

**Figure 14 ijms-24-16555-f014:**
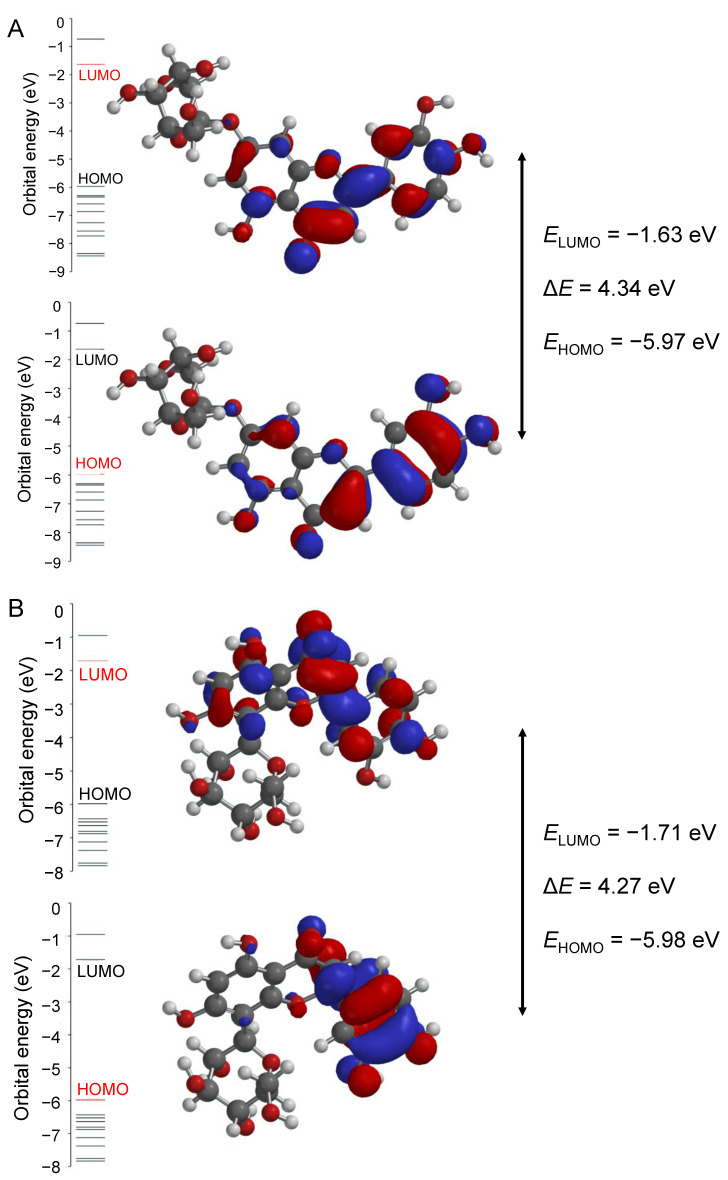
The diagram of energy of the Highest Occupied Molecular Orbital (HOMO) and Lowest Unoccupied Molecular Orbital (LUMO) and the energy difference between the HOMO and LUMO orbitals (Δ*E*) for 7-Lut (**A**) and 8-Lut (**B**), predicted with the Density Functional Theory (DFT) and Becke’s three parameter hybrid exchange functional with the Lee–Yang–Parr correlation functional (B3LYP)-6-311(d,p) method.

**Figure 15 ijms-24-16555-f015:**
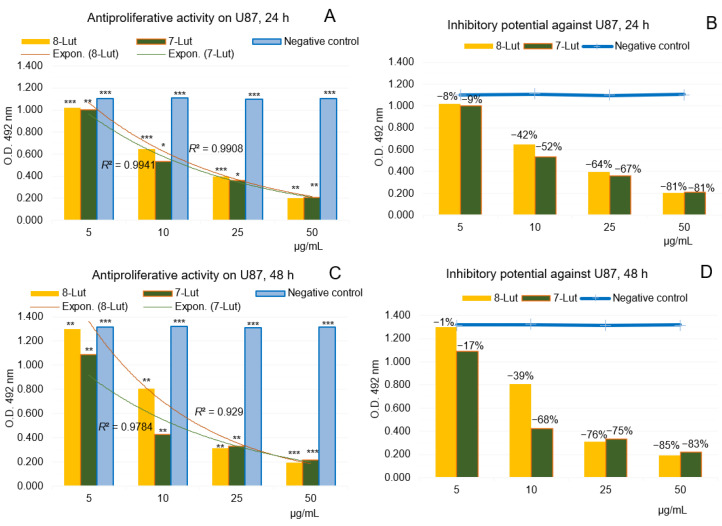
Anti-proliferative potential of 7-Lut and 8-Lut tested on human tumor glial cell line U87, by MTS assay. (**A**,**B**) cell viability after 24 h of exposure to 7-Lut and 8-Lut. (**C**,**D**) cell viability after 48 h of exposure to 7-Lut and 8-Lut. (*n* = 3). Notation (*) means results without statistical significance (*p* > 0.05); (**) means results with statistical significance (0.05 < *p* < 0.01); (***) means results with high statistical significance (*p* < 0.01); 7-Lut means luteolin-7-*O*-glucoside; 8-Lut means luteolin-8-*C*-glucoside; MTS means 3-(4,5-dimethyl-thiazol-2-yl)-5-(3-carboxy-methoxy- phenyl)-2 -(4-sulfophenyl)-2H-tetrazolium (MTS).

**Figure 16 ijms-24-16555-f016:**
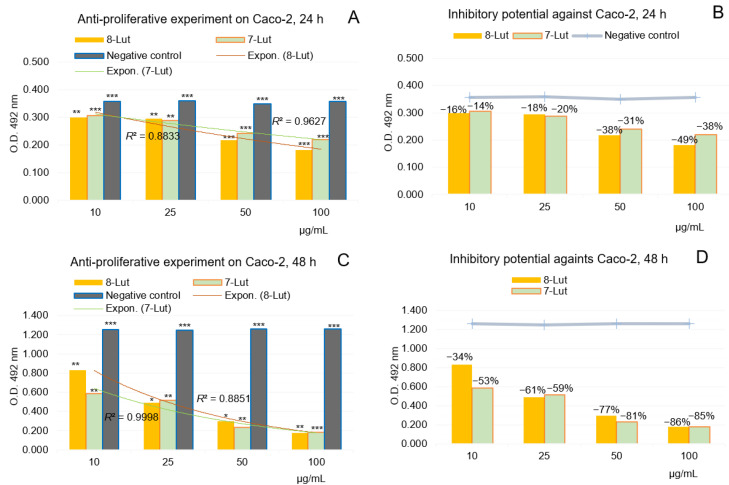
Anti-proliferative potential of 7-Lut and 8-Lut tested on human tumor colon cell line Caco-2, by MTS assay. (**A**,**B**) cell viability after 24 h of exposure to 7-Lut and 8-Lut. (**C**,**D**) cell viability after 48 h of exposure to 7-Lut and 8-Lut. (*n* = 3). Notation (*) means results without statistical significance (*p* > 0.05); (**) means results with statistical significance (0.05 < *p* < 0.01); (***) means results with high statistical significance (*p* < 0.01); 7-Lut means luteolin-7-*O*-glucoside; 8-Lut means luteolin-8-*C*-glucoside; MTS means 3-(4,5-dimethyl-thiazol-2-yl)-5-(3-carboxy- methoxyphenyl)-2 -(4-sulfophenyl)-2H-tetrazolium (MTS).

**Figure 17 ijms-24-16555-f017:**
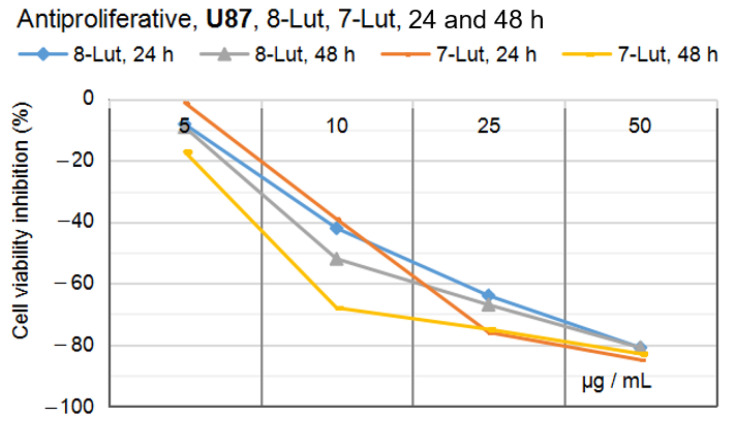
The viability response of U87 cells to 7-Lut and 8-Lut after 24 and 48 h of exposure.

**Figure 18 ijms-24-16555-f018:**
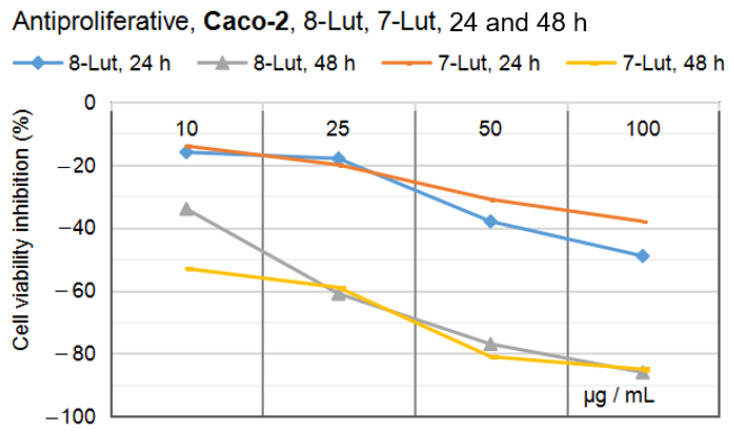
The viability response of Caco-2 cells to 7-Lut and 8-Lut after 24 and 48 h of exposure.

**Figure 19 ijms-24-16555-f019:**
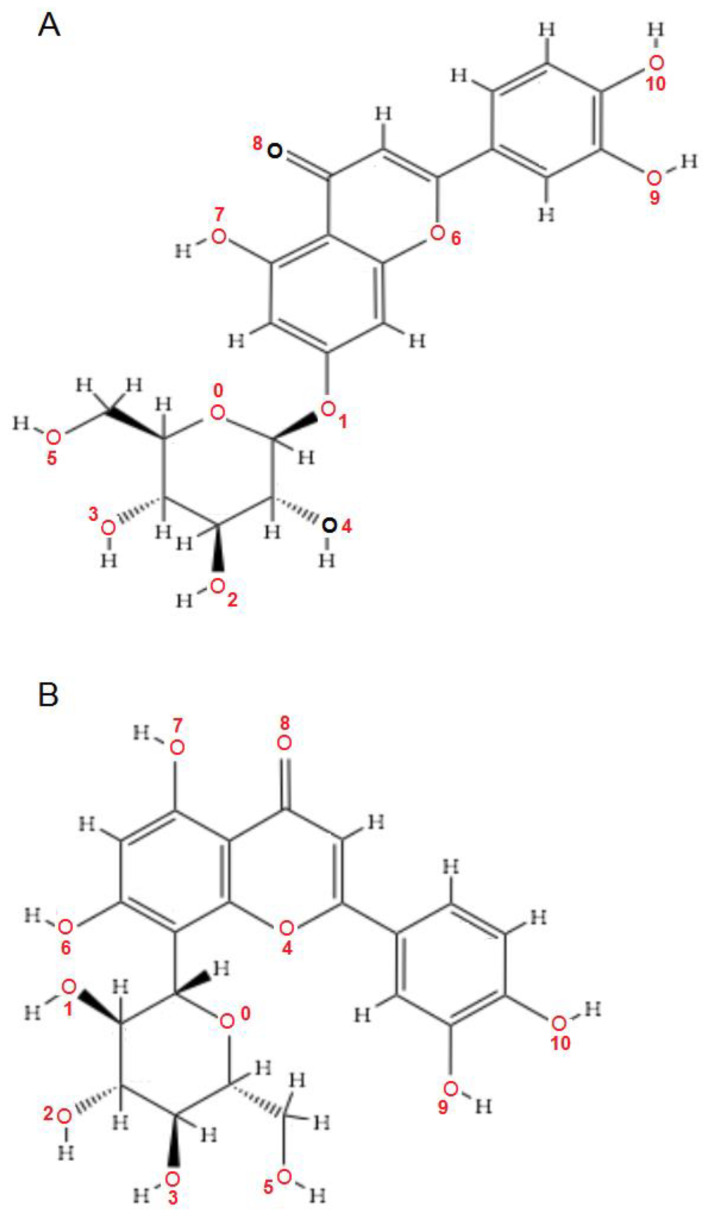
2D structures of luteolin-7-*O*-glucoside (**A**) and luteolin-8-*C*-glucoside (**B**).

**Table 3 ijms-24-16555-t003:** Calculated quantum-chemical parameters of the three test phenolics.

Quantum Parameter	7-Lut, Cynaroside	Luteolin, Aglycone	8-Lut, Orientin
*E* _HOMO_	−5.97	−6.05	−5.98
*E* _LUMO_	−1.63	−1.84	−1.71
Δ*E*	4.34	4.21	4.27
*I*	5.97	6.05	5.98
*A*	1.63	1.84	1.71
*χ*	3.80	3.94	3.84
*η*	2.17	2.105	2135
*σ*	0.46	0.47	0.47
*μ*	−3.80	−3.94	−3.84
*ω*	3.33	3.68	3.45

Where *E*_HOMO_ represents the energy of the Highest Occupied Molecular Orbital (HOMO); *E*_LUMO_ is the energy of the Lowest Unoccupied Molecular Orbital (LUMO); Δ*E* refers to the difference between the bonding (HOMO) and anti-bonding (LUMO) orbitals; *I* represents the ionization potential (−*E*_HOMO_); *A* is the electron affinity (−*E*_LUMO_); *χ* stands for electronegativity, given by the formula *(I + A)/2*; *η* is the global hardness (*I* − *A*)/2; *σ* is the local softness, given by the 1/*η* ratio; *μ* is the chemical potential, *μ* = −*χ* = −(*E*_HOMO_ + *E*_LUMO_)/2; *ω* represents the global electrophilicity index given by the relation *μ^2^/2η*; all parameters are expressed in eV.

**Table 4 ijms-24-16555-t004:** Predicted drug-likeness parameters of 7-Lut, 8-Lut, and curcumin.

Test Compounds	Atoms	Weight (Daltons)	Flexible Bonds	Lipinski Violations	Hydrogen Donors	Hydrogen Acceptors	logP
Curcumin	49	370.40	9	0	2	6	3.00
7-Lut	52	448.38	4	2	7	11	1.57
8-Lut	52	448.38	3	2	8	11	−0.70

logP stands for the water-octanol partition coefficient.

## Data Availability

Data are contained within the article.

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
