# Peer review of "Anti-Proliferative Potential of Cynaroside and Orientin—In Silico (DYRK2) and In Vitro (U87 and Caco-2) Studies"

_ijms, 2023, doi:10.3390/ijms242316555_

Round 1

Reviewer 1 Report

Comments and Suggestions for Authors

Comments on the Quality of English Language

Author Response

Esteemed Reviewers, 

Esteemed Editorial Board,

 IJMS - MDPI

 Ref.: Manuscript ID: ijms-2674063; Title: Antiproliferative potential of cynaroside and orientin; in silico and in vitro studies on DYRK2, and U87 and Caco-2. Authors: Lucia Camelia Pirvu *, Lucia PINTILIE, Adrian ALBULESCU, Amalia STEFANIU, Georgeta Neagu *

 First of all, the authors thank you very much for your very careful review of our paper! 

The reviewers' comments are actually a great opportunity to interact with specialists from related domains and better understand and present your studies. Based on your comments and suggestions, the paper has been now significantly improved.

The article has been entirely rebuilt in the INTRODUCTION, DISCUSSIONS and CONCLUSIONS sections; also, the article has been English editing, typographical errors and tables numbering corrected; the changes are marked with blue color, while red color is for new introduced information.

Reviewer 1:

Starting from the title the authors did not perform in vitro study on the DYRK2 gene (only in silico study), so please correct the title to represent the study correctly – REVISED;

  1. Extensive English editing is needed – REVISED;
  2. Typographical errors were observed starting from the abstract such as; missed spaces, double dots, and .[ref] or [ref]. you should adjust these – REVISED;

4/5. In the introduction section, the paragraph mentioned ‘’ Of the numerous formulation techniques available, the current emphasis is on encapsulating …...’’ has been deleted and replaced with <suggestive quantum calculations and docking results for introducing new applications of the existing natural compounds> - the bibliography indicated is very useful for our knowledge and for future studies in this subject – REVISED;

  1. The titles of the tables should be revised, you type the tables title twice as in table 4 – REVISED;
  2. In the discussion section, why is the table 3 is inserted, this is not a review, you designed the manuscript on 7-lut and 8-lut if you insist to place this table in the discussion, you have to do docking on each compound in the table – REVISED;

- Esteemed reviewer, we agree with the observation, but we could not give up the information from this table (wrongly marked with 3), so we made a new paragraph with the content of the two test compounds (or they generic series) in the common vegetal food;

  1. Be sure of tables numbering you named both tables (table 3) – REVISED;
  2. You have to insert the figures of (HOMO) and (LUMO) of the two compounds and the softness value in a high-resolution figure – REVISED.

We hope that both, the Reviewers and the Editor in Chief will agree with our answers and revisions made in the paper, and will accept the publication of the work in this improved form.

 Sincerely yours,

Lucia Pirvu et al. 

Reviewer 2 Report

Comments and Suggestions for Authors

The authors present the antiproliferative effect of two luteolin glycosides using in silico and in vitro experiments. The referee has the following comments and questions regarding the results.

1. What was the reason for choosing only the two compounds in question, after countless similar natural molecules could have been investigated. So many natural substances have been isolated recently that more than a few of Aldrich's compounds should not have been touched upon.

2. It is a fundamental requirement that the compounds in the paper be presented with clear structural formulas from an organic chemistry point of view. Modeled linear formulas make it difficult to understand docking connection options.

3. IC50 values are not very informative by themselves. In order to evaluate these results, the corresponding values of a known control used in therapy should be presented.

4. Overall, two substances on two cell lines, moreover, without a control, cannot be considered relevant.

Author Response

Esteemed Reviewers, 

Esteemed Editorial Board,

 IJMS - MDPI

 Ref.: Manuscript ID: ijms-2674063; Title: Antiproliferative potential of cynaroside and orientin; in silico and in vitro studies on DYRK2, and U87 and Caco-2. Authors: Lucia Camelia Pirvu *, Lucia PINTILIE, Adrian ALBULESCU, Amalia STEFANIU, Georgeta Neagu *

First of all, the authors thank you very much for your very careful review of our paper! 

The reviewers' comments are actually a great opportunity to interact with specialists from related domains and better understand and present your studies. Based on your comments and suggestions, the paper has been now significantly improved.

The article has been entirely rebuilt in the INTRODUCTION, DISCUSSIONS and CONCLUSIONS sections; also, the article has been English editing, typographical errors and tables numbering corrected; the changes are marked with blue color, while red color is for new introduced information.

Reviewer 2:

  1. What was the reason for choosing only the two compounds in question, after countless similar natural molecules could have been investigated. So many natural substances have been isolated recently that more than a few of Aldrich's compounds should not have been touched upon. – REVISED;

- Esteemed reviewer, we indeed failed to explain the essence of the selection of 7-Lut and 8-Lut in the present study.  In short, no matter how sophisticated the new discovered compounds are, they are all O-glycosylated or C-glycosylated derivatives, and from the point of view of their metabolism in humans, they finally reach as two forms available for the circulatory system: aglycone/luteolin if they come from O-glycosides and luteolin-8-C-monoglycoside if they come from 8-C-glycosides; this is because 6-C-glycosides have been shown to be metabolized by microbiota. However, the situation may change as new discoveries are made, or compounds with exceptional activities will be revealed by in silico and in vitro studies. However, they must be delivered encapsulated, exactly to avoid their typical metabolism in humans that equalizes all compounds at these two active forms.

  1. It is a fundamental requirement that the compounds in the paper be presented with clear structural formulas from an organic chemistry point of view. Modeled linear formulas make it difficult to understand docking connection options. – REVISED;
  2. IC50 values are not very informative by themselves. In order to evaluate these results, the corresponding values of a known control used in therapy should be presented.

- Even if it is not fully satisfactory, IC50 is a universal language that suggests the degree of the efficiency of test products under similar test conditions. In the case of antitumor products, based on the IC50 values, an estimation can be made of whether the respective compounds can be used as direct treatment, as preventive treatment, or as treatment after the elimination of the cancer tissue by generic treatment. In the present case, considering the punctual IC50 values of 8-Lut and 7-Lut on the two cell lines, U87 an dCaco-2, situated in the range 25 - 100 yg/mL, the comparison with a chemical drug used in current therapy has no point; it is known that usually they have IC50 < 1 µg/mL. Excepting only few vegetal compounds, the value of the plant compounds generally comes from their collateral antitumor effects, therefore they are effective for preventive purpose or for enhancing the human body capacity to fight with the cancer recurrence after the tumor eradication by chemical treatment. Even more if the IC50 values of the compounds in the cytotoxicity experiment are much higher than their effective concentration in the anti-proliferative experiment.

  1. Overall, two substances on two cell lines, moreover, without a control, cannot be considered relevant.

- The relevance of the study is in the context in which the two compounds’ are in fact the basic model of most of luteolin compounds subjected to the metabolism in humans; we will continue studies on other human tumor cell lines in the near future, as we receive new funding.

We hope that both, the Reviewers and the Editor in Chief will agree with our answers and revisions made in the paper, and will accept the publication of the work in this improved form.

 Sincerely yours,

Lucia Pirvu et al. 

Round 2

Reviewer 1 Report

Comments and Suggestions for Authors

The manuscript is improved

Comments on the Quality of English Language

minor english editing is required, some typographical errors are noticed

Author Response

Esteemed Professor, the article has been once again revised and corrected. The authors thank you very much for the valuable indications during the revision process.

Reviewer 2 Report

Comments and Suggestions for Authors

After studying the paper and the answers of the authors, I can say that all in all, the authors answered the referee's comments in an acceptable way, the paper has been properly revised, so I can recommend its acceptance.

Author Response

(The authors gave the same response as above.)
